# Site-Specific Seismic Analysis of Buildings Supported by Lightly Reinforced Precast Concrete Walls

**Xiangzhe Weng** [1,*] **, Ryan D. Hoult** [2] **and Elisa Lumantarna** [1]

1 Department of Infrastructure Engineering, The University of Melbourne, Parkville, VIC 3010, Australia
2 Institute of Mechanics, Materials and Civil Engineering, Université catholique de Louvain, 1348 Louvain-la-Neuve, Belgium
* Correspondence: xiangzhew@student.unimelb.edu.au; Tel.: +61-4-5016-2060

**Abstract:** This paper aims to show the application of site-specific response spectra in the analysis of buildings that are supported by lightly reinforced precast concrete walls. Previous surveys on load-bearing precast reinforced concrete walls in multi-storey buildings in low-to-moderate seismic regions have found that many existing precast walls are lightly reinforced with a connection reinforcement ratio less than the wall reinforcement ratio. When these precast walls are subjected to reversed cyclic loads, the lateral response is typically controlled by rocking and the ultimate performance is governed by the ruptures of connection dowels. This paper uses moment–curvature analyses in combination with plastic hinge analyses to evaluate the force–displacement capacity of planar lightly reinforced load-bearing precast walls. The seismic performance of a building supported by these lightly reinforced precast walls can then be assessed by superimposing the capacity curve and the inelastic site-specific response spectra developed for the building site. The proposed analytical approach is illustrated through a case study building. By comparing a lightly reinforced precast wall with a comparable limited ductile reinforced concrete wall, it is also found that, although these two walls exhibit similar force capacities, the ultimate displacement capacity of the lightly reinforced precast wall is significantly lower. This finding highlights the potential seismic vulnerability of lightly reinforced precast walls in some existing buildings.

**Keywords:** site-specific response spectra; low-to-moderate seismicity; precast walls; grouted dowels; capacity spectrum method; moment–curvature analysis; plastic hinge analysis; non-linear static





## 1. Introduction

In the seismic design and analysis of a building structure, design loading codes typically require engineers to use a code response spectrum based on the soil types at the building site. However, previous research has identified limitations in the design response spectra specified in some codes (e.g., AS 1170.4 [1]), such as misrepresentation of the site amplification effect [2–4]. In order to better predict ground motion in a specific building site, researchers have recommended using site-specific response spectra for the earthquake-resistant design of buildings (i.e., site-specific seismic design) [3,5–9]. For example, Hu et al. [3] elaborate on the principles and steps to derive the site-specific response spectrum for a given site. Khatiwada et al. [6] adopt a similar approach to developing site-specific response spectra based on AS 1170.4 [1] for case study buildings located in Australia for a return period of 2500 years. Khatiwada et al. [6] also compare proposed site-specific response spectra with the code response spectrum to highlight the benefit of using the site-specific design approach. The approach for deriving site-specific response spectra proposed in these studies [3,5,6] is applicable to any stable continental region. Although site-specific seismic design and analysis have been widely discussed by researchers worldwide, this approach is rarely used by practitioners in the seismic evaluation and design of reinforced concrete (RC) structures, particularly in low-to-moderate seismic regions. One reason is the

lack of guidance available in the industry. Within the limited existing studies focusing on site-specific seismic design and analysis, there is a paucity of research demonstrating how to adopt site-specific response spectra for assessing and designing precast RC walls.

The use of precast RC walls as primary load-bearing structural elements in single- and multi-storey buildings is becoming popular in low-to-moderate seismic regions, such as Australia. Compared to traditional cast-in-place construction, precast construction has several advantages, including reduced construction time and labour costs, as well as reduced waste [10]. The solid RC precast panel is widely adopted in regions of low-to-moderate seismicity, which uses grouted dowels for the horizontal connections and welded stitch plates or in-situ wet joints as the vertical connections [10–12]. The design and detailing of these precast systems aim to attain the equivalent structural behaviour of cast-in-situ RC walls. Nevertheless, limited experimental tests on existing precast walls in low-to-moderate seismicity regions have recently identified vulnerability in these types of walls [13,14]. Due to the poor design and detailing practices adopted in some existing precast walls, these structural elements could behave significantly differently from the behaviour of well-detailed cast-in-situ RC walls under strong ground motions. Figure 1 illustrates two examples of typical detailing in lightly reinforced precast walls in Australia [12] (note that the ducts and connection dowels are not illustrated in Figure 1). One of the leading reasons for the abundant use of poorly detailed lightly reinforced precast walls in the building stock in low-to-moderate seismic regions, such as Australia, is that there is a paucity of experimental and numerical investigations focusing on their seismic performance, with some efforts to rectify this recently (e.g., [13,14]).

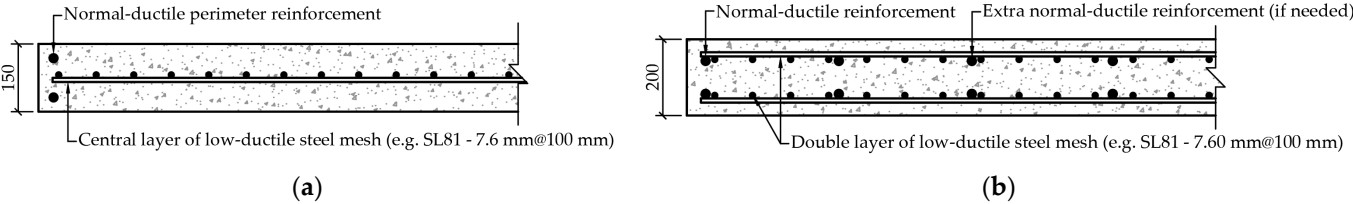

**Figure 1.** Two examples of reinforcement detailing in lightly reinforced precast panels typically found in Australia (corrugated ducts and connection reinforcement not shown). (**a**) Centrally reinforced precast wall. (**b**) Doubly reinforced precast wall.

Recent studies on existing load-bearing precast walls in Australia and New Zealand (e.g., [10–15]) have identified some crucial issues that need to be resolved before there is widespread adoption of such structural elements in the building stock of low-, mid-, and even high-rise structures. Firstly, past design considerations of precast walls in low-to-moderate seismicity regions were widely assumed to be 'gravity load-resisting only', while assuming that all the lateral force would be resisted by other cast-in-place cores in buildings, even though these precast walls are effectively tied by rigid floor diaphragms [15]. This false design assumption ignores the potential lateral loads being distributed to precast walls depending on their lateral stiffness. As a result, wind and seismic actions on these structural elements might not be accurately considered in the structural design process. In fact, even if the seismic action on precast walls was assessed by engineers in the past, the previous AS 3600:2009 Australian concrete structures code [16] allowed for a 'simplified design' approach, which was based on stress at the wall boundaries [17]. The second critical issue relates to the amount of longitudinal reinforcement detailing in these elements, where existing precast walls and even some newly designed precast walls [10] are only reinforced with one, sometimes two layers of low-ductile steel mesh (i.e., 'L-type' in Australia). In some limited cases, discrete normal-ductile (i.e., 'N-type' in Australia) deformed steel bars can be placed at the two boundaries of the precast walls and between mesh reinforcement, as shown in Figure 1b. Some experimental tests of RC walls detailed with low-ductile steel have shown very limited displacement capacity and are typically governed by early rupture of the reinforcement [18,19]. Even in cases where the wall

boundaries are reinforced with normal-ductile rebars (e.g., Figure 1), it is possible that the low-ductile mesh would still experience premature fracturing at low displacement demand. Prior to the heavily revised AS 3600 Australian concrete structures standard [20] in 2018, a low steel reinforcement content would typically be adopted for the connection at the base of the wall panel in the grouted dowels [13]. This low reinforcement amount results in the formation of a single, primary crack at the base [14,21,22]; inelastic strains are likely to concentrate at the connection dowels that cross this base crack. As a result, such precast walls will be primarily subjected to rocking deformation with little flexural deformation [14], particularly when the axial load ratio of the precast walls is small (e.g., an axial load ratio of approximately 5% in [13,14]). Sliding at the wall base (i.e., the shear friction behaviour) only becomes noticeable at large drift levels when the precast walls approach failure [14]. The ultimate contribution of sliding shear to the total lateral deformation of the lightly reinforced precast walls is still less than 10% in the majority of tested specimens in Seifi et al. [14]. In particular, the sliding deformation becomes marginal if the walls are subjected to additional axial loads [14]. The shear deformation of lightly reinforced precast walls is also negligible relative to the overall lateral drift of the walls tested in Seifi et al. [14]. The development of rocking deformation at the wall base is mainly due to bond slips between connection dowels and grout, as well as elongation of the connection dowels. The failure of precast walls that exhibit extensive rocking deformation is likely to be caused by sudden fracturing (or rupture) of the connection dowels, which could significantly reduce the lateral stability of the precast walls [13]. The rocking deformation can also cause displacement incompatibility between the precast walls and other structural elements connected to the walls [23]. Additionally, the inelastic deformation and sudden rupture of dowels are hard to predict and irrecoverable, which can pose difficulties in post-earthquake assessment and rehabilitation.

As a response to these seismic deficiencies of lightly reinforced load-bearing precast walls identified in recent experimental investigations [13,14], structural design codes in different countries have started reviewing and updating the relevant provisions for precast structural walls. For example, as mentioned briefly in the previous paragraph, the latest-generation Australian concrete structures code AS 3600:2018 [20] has made several updates on the detailing requirements for precast structural walls. In particular, the AS 3600:2018 [20] code requires that 'limited-ductile' precast walls, with an assigned ductility factor ($\mu$) of 2, should have a reinforcement ratio of the connection dowels exceeding the reinforcement ratio of vertical steel bars in the wall panel. If this requirement is not met, the wall is considered to be non-ductile (e.g., $\mu$ = 1), and design checks are only provided for full elastic earthquake actions. The AS 3600:2018 [20] also prohibits the use of low-ductile mesh and a single layer of reinforcement in 'limited-ductile' precast walls. Furthermore, even if the precast walls are primarily used for gravity-load-resisting purposes in buildings, the AS 3600:2018 [20] still recommends considering lateral earthquake loads acting on the walls based on their in-plane lateral stiffness. It can be expected that if the enhanced design and detailing practices are adopted, the seismic behaviour of precast RC walls could be comparable to that of monolithic cast-in-place walls, achieving similar limited-ductile flexural action and a better distribution of cracks up the wall height. However, since lightly reinforced load-bearing precast walls were widely used in the building and construction industry prior to the implementation of a more robust design code (e.g., AS 3600:2018), it is necessary to examine the seismic performance of these poorly detailed precast structural elements.

This paper provides guidance for assessing the seismic performance of planar (i.e., rectangular) lightly reinforced precast walls in regions of low-to-moderate seismicity using the capacity spectrum method (CSM). The CSM combines the capacity curve obtained from a simplified non-linear static analysis (i.e., push-over analysis) using a plastic hinge analysis and the site-specific response spectra demand curves. It is worth emphasising that the term 'lightly reinforced' for the precast walls examined here refers to the minimum reinforcement detailing according to the AS 3600:2009 [16], prior to the adoption of more stringent design guidelines in the AS 3600:2018 [20]. Some of these 'poor' detailing practices were also

observed in other countries (e.g., [11]). Section 2 of this paper elaborates on the concepts and steps involved in using the proposed analytical approach in the seismic evaluation of buildings supported by precast RC walls. Section 3 applies the proposed method to assess the force–displacement response of a case study building supported by four rectangular lightly reinforced precast walls. By superimposing the capacity curve of the case study building and the site-specific response spectra in acceleration and displacement (e.g., *RSA* vs. *RSD*) format, the potential vulnerability of lightly reinforced precast walls with dowelled connections is highlighted. The force–displacement behaviour of these precast walls is further compared with that of a conventional rectangular cast-in-place RC wall using the simplified analytical equations in Khatiwada et al. [6]. The seismic assessment method proposed in this paper is applicable to lightly reinforced precast walls in other stable continental regions, provided that the detailing of the precast walls is similar to the one discussed in this paper.

## 2. Guidance for Site-Specific Seismic Analysis of Precast RC Walls

### 2.1. Simplified Push-Over Analysis

A force-based approach is commonly employed in the seismic design and evaluation of both cast-in-situ and precast building structures. Seismic design standards in most countries specify two force-based methods, namely, equivalent static force analysis and linear dynamic modal analysis, to design and assess buildings. As required by most design codes, practising engineers can choose one of the two methods mainly based on the importance level, sub-soil class and height of the buildings [1]. These force-based approaches aim to provide building structures with sufficient strength such that, under 'rare' earthquake ground motions, satisfactory performance is achieved (e.g., 'life safe' or 'collapse prevention'). The concept of the force-based approach is based on the 'equal-displacement' assumption [24,25], as illustrated in Figure 2. Structures are allowed to be designed for reduced strength ($F_d$), which is obtained by scaling down the full elastic earthquake force ($F_e$) obtained from an elastic response spectrum by the prescriptive ductility factor ($\mu$) and overstrength factor ($\Omega$) (i.e., $\Omega = 1/S_p$, where $S_p$ is defined in [1]). The detailing of structural elements should conform with the relevant requirements prescribed in the material design codes. Although the force-based approach has been used internationally for decades, researchers have identified several constraints and issues with adopting this approach in seismic design and assessment for RC structures [24]. In particular, the force-based approach specified in most international design codes is not directly associated with structural damage and collapse, which are mainly controlled by material strains and the displacement of structures, rather than lateral forces [24,26]. Furthermore, the ductility factor used in the force-based approach often does not represent actual displacement ductility demands on structural systems [24].

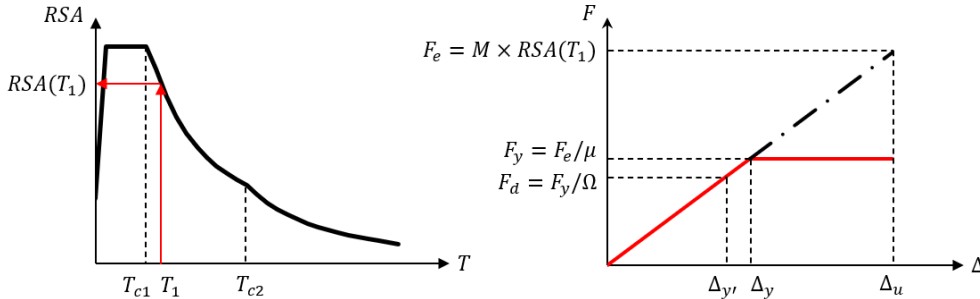

**Figure 2.** Force-based seismic design approach.

In order to overcome the issues of the force-based approach, a displacement-based seismic design and evaluation approach has been proposed by researchers. One of the most comprehensive explanations of the framework of displacement-based seismic design was provided by Priestley et al. [24]. The main goal of the displacement-based design

approach is to provide building structures with sufficient displacement capacity. In this approach, structural inelasticity is explicitly assessed in the seismic analysis by either non-linear static analysis (e.g., push-over analysis) or non-linear dynamic analysis (e.g., time-history analysis). Selection of the proper analytical tool mainly depends on the complexity, irregularity and significance of the structure and requires the judgement of engineers [27]. Although non-linear time-history analysis tends to offer the most accurate prediction of structural behaviour, it is rarely used by practitioners due to its sophistication and enormous demands on computational time, as well as site and building information. By contrast, non-linear push-over analysis is less time-consuming and information-demanding and has been found to be reasonably reliable in predicting the lateral drift behaviour of RC structures in regular low-to-medium-rise buildings [26,28–32].

A push-over analysis provides the global force–displacement capacity curve of a structure, which can then be superimposed with the inelastic acceleration–displacement demand curves, as illustrated in Figure 3. Such superposition is frequently referred to as the capacity spectrum method (CSM) in past research studies [26–28,31–34]. The simplicity of the CSM offers practising engineers a ready and time-efficient way to evaluate the seismic performance of RC buildings using a displacement-based design concept. This paper elaborates on the principles of and procedures for using this method in the seismic analysis of multi-storey buildings supported by lightly reinforced precast walls. Khatiwada et al. [6] adopted a similar approach in analysing the seismic performance of buildings supported by limited-ductile cast-in-situ walls. However, due to the difference in detailing practices between well-detailed cast-in-situ walls and lightly reinforced precast walls, the details of using the CSM for lightly reinforced precast walls are highlighted in this paper.

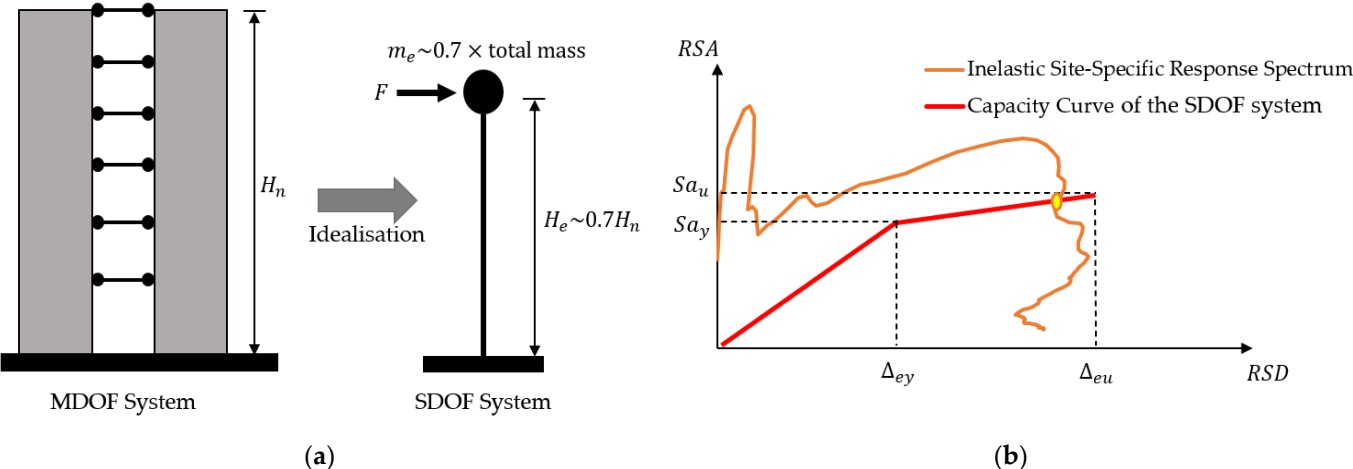

**Figure 3.** Displacement-based capacity spectrum method. (**a**) SDOF idealisation. (**b**) Capacity and response spectrum curves.

As illustrated in Figure 3, the capacity spectrum method discussed in this paper includes four steps:

- Step 1: Idealisation of a multi-degree-of-freedom (MDOF) wall building to an equivalent single-degree-of-freedom (SDOF) system. The effective mass ($m_e$) and height ($H_e$) of the equivalent SDOF system can be estimated using Equations (1) and (2), assuming that the deflection shape of the building is triangular:

$$H_e = \frac{\sum\left(m_i h_i^2\right)}{\sum(m_i h_i)} \tag{1}$$

$$m_e = \frac{\sum h_i m_i}{H_e} \tag{2}$$

where $m_i$ is the mass of the *i*th floor of the building; $h_i$ is the height of the *i*th floor above the foundation. Alternatively, the effective mass of the SDOF system can be estimated to be 70% of the total mass of the multi-storey wall building [24]. The effective height of the SDOF system is also commonly approximated as 70% of the total height ($H_n$) of the building [24]. The structural period of the SDOF system can be assumed to be equal to the fundamental period of the MDOF building, with the same damping properties [35]. Therefore, the CSM adopted in this paper is more applicable to low-to-medium-rise buildings (e.g., less than seven storeys [30]) without apparent in-plane or vertical irregularities (i.e., higher mode effects, as well as torsion, are not considered for these analyses).

- Step 2: Derivation of the capacity curve of the equivalent SDOF system, which represents the global seismic performance of a complex MDOF building structure. The capacity curve is typically presented in a bi-linear format with two controlling points: the nominal yield point and the ultimate point. For regular low-to-medium-rise buildings supported by lightly reinforced precast walls, sectional-based moment–curvature analysis (or fibre-element analysis) and plastic hinge analysis can be used to determine the bi-linear force–displacement capacity curve. Section 2.2 of this paper summarises the procedures for conducting moment–curvature analysis for the critical section of lightly reinforced precast walls and then calculating the force–displacement response using plastic hinge analysis, as provided in Section 2.3.

- Step 3: Derivation of the site-specific response spectra (i.e., demand curves). The discussion on the derivation of the site-specific response spectra is out of the scope of this paper. Readers can refer to Hu et al. [3] and Khatiwada et al. [6] for more details. As a demonstration, Section 3 of this paper applies the site-specific response spectra developed by Khatiwada et al. [6] to the seismic analysis of a case study building supported by lightly reinforced precast walls. It is noted that since the capacity curve represents the inelastic behaviour of structures, elastic demand curves should also be converted to inelastic curves using Equations (3)–(5) from Fajfar [36], as illustrated in Figure 4. The value of the overstrength factor ($\Omega$) can be determined as per a seismic loading code (e.g., [1]) by assuming that the lightly reinforced precast walls are non-ductile. The ductility factor ($\mu$) is the ratio of the ultimate displacement ($\Delta_{eu}$) and the yield displacement ($\Delta_{ey}$), as shown in Figure 3b, obtained from the capacity curve. The corner period ($T_{c1}$) is the period at the intersection of the acceleration-controlled region and the velocity-controlled region in an elastic code response spectrum (i.e., the constraining period of the acceleration-controlled region) [36].

$$RSA_{inelastic} = \frac{RSA_{elastic}}{R_\mu \Omega} \tag{3}$$

$$RSD_{inelastic} = \frac{\mu}{R_\mu} RSA_{elastic} \left( \frac{T}{2\pi} \right)^2 \tag{4}$$

$$R_\mu = (\mu - 1)\frac{T}{T_{c1}} + 1 \leq \mu \tag{5}$$

- Step 4: Superimposition of the capacity curve and demand curves: if the ultimate point of the capacity curve exceeds the envelope of the demand curves, the seismic performance of the examined precast wall is satisfactory. Otherwise, the structural elements are vulnerable to seismic ground motions. Additionally, intersections of the capacity and demand curves are often called 'performance points', which can be used to further optimise the design solution.

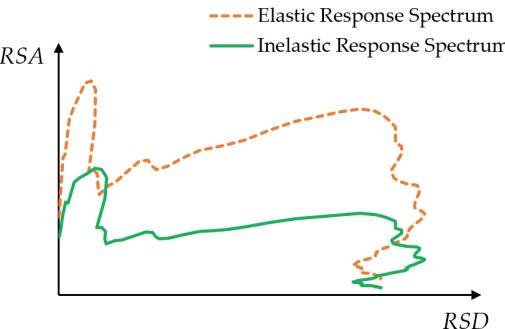

**Figure 4.** Illustration of the inelastic site-specific response spectrum.

### 2.2. Moment–Curvature Analysis of Lightly Reinforced Precast Walls

Moment–curvature analysis (also known as sectional analysis) has been widely used to assess the capacities of RC structural elements [30,37,38]. There is a range of freely accessible programs developed by different researchers for running sectional-based moment–curvature analyses of RC elements, such as the Microsoft Excel-based program WHAM [39–41]. This paper will use WHAM, which is free to download (https://downloads.menegon.com.au/ (accessed on 29 December 2022)) [41]. The reliability of WHAM has been validated [40]. The moment–curvature analysis adopted in this paper is proposed based on the fibre-element analysis used in past research studies [35,40,42,43]. The main steps for undertaking non-linear sectional-based moment–curvature analyses are summarised below, with a demonstration analysing two experimental RC precast wall units, Wall 1 (SW1) and Wall 4 (SW4) [14]. Illustrations and summaries of the geometry and reinforcing steel of the two wall units are given in Figure 5 and Table 1, respectively. The longitudinal reinforcement ratio ($\rho_v$) and connection reinforcement ratio ($\rho_{cd}$) of SW1 and SW4 are also provided in the table. The failure mechanism of the two wall units under in-plane reversed-cyclic loading is the fracturing of connection dowels at the single crack formed at the wall–foundation interface. The focus of this paper is to explain the concepts of undertaking a sectional analysis for rectangular lightly reinforced precast walls, rather than the use of the software (e.g., WHAM). Readers can refer to Menegon [39] and Menegon et al. [40] for instructions on using the software. In this paper, the vertical connections between precast panels (e.g., welded stitch plates or wet joints) are assumed to be rigid enough to develop full composite action for the connected walls. The evaluation of these vertical connections is outside the scope of this paper. It is noted that wet joints typically have equivalent strength and stiffness to a cast-in-place construction. In addition, for welded stitch plates, the rigidity of the connections depends on their configurations. Menegon et al. [44] developed mathematical models to predict the stiffness of various welded stitch plates. For demonstration purposes, the calculations of material properties for the moment–curvature analyses in the following sections are mainly based on Australian standards. However, readers can adopt the same methodology with material properties measured experimentally or specified in other design standards to assess similar lightly reinforced precast walls in other regions and countries. These material properties and even material models can typically be user-defined in software (e.g., WHAM [41]).

Step 1—Identifying the critical cross-section: theoretical moment–curvature analysis is based on analysis of the critical cross-sections of lightly reinforced precast walls. Hence, the first step is to identify the critical cross-sections and then define their dimensional and reinforcement properties. For this research, it is assumed that the connection reinforcement ratio is less than the longitudinal reinforcement ratio of the wall panel. As explained previously, in this case, a single primary crack is likely to form at the wall base (i.e., the grout bedding layer), with the seismic performance of the wall controlled by the concentration of inelastic strains from the dowel connections [14]. As such, the critical section of a lightly reinforced precast wall panel consists only of the connection dowels, where the reinforcement embedded in the wall panel remains essentially elastic, as shown in Figure 6. This behaviour, corresponding to a single crack causing the fracturing of

connection dowels, was observed experimentally during the testing of the two wall units studied here (i.e., SW1 and SW4 [14]).

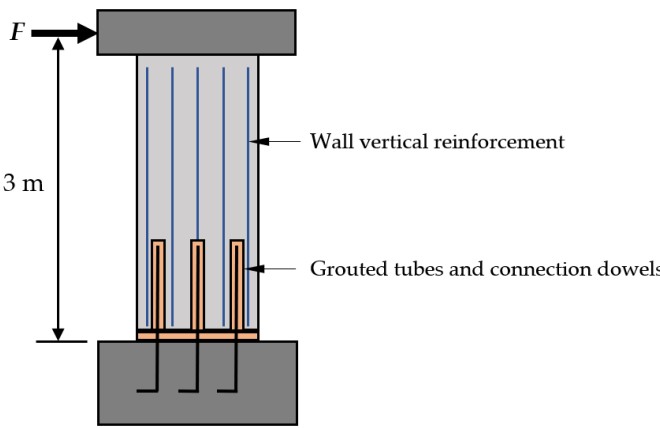

**Figure 5.** Illustration of SW1 and SW4 [14].

**Table 1.** Summary of two precast wall units SW1 and SW4 [14].

| Specimens | Dimension (Length × Height × Thickness) | Connection Dowel [1] | Wall Vertical Reinforcement [1] |
|---|---|---|---|
| SW1 | 1000 × 3000 × 150 | 16@400 ($\rho_{cd}$ = 0.40%) (100 mm cover to dowel centre) | Single layer 12@225 ($\rho_v$ = 0.38%) (50 mm cover to rebar centre) |
| SW4 | 1000 × 3000 × 200 | 16@400 ($\rho_{cd}$ = 0.30%) (100 mm cover to dowel centre) | Double layer 12@225 ($\rho_v$ = 0.57%) (50 mm cover to rebar centre) |

[1] 16@400 means a bar diameter of 16 mm and a spacing of 400 mm. Similarly, 12@255 means a bar diameter of 12 mm and a spacing of 225 mm.

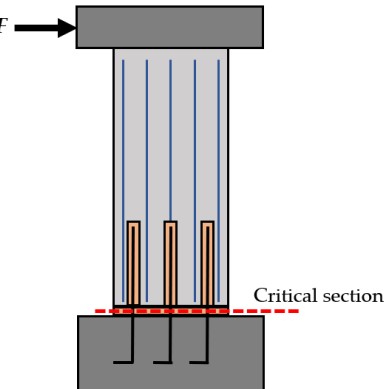

**Figure 6.** Potential critical cross-section of rectangular lightly reinforced precast walls.

Step 2—Modelling the critical cross-section: the geometry of the cross-section and the locations of the vertical reinforcement across the section should be accurately modelled according to structural drawings or the reinforcement layout in existing precast walls, as illustrated in Figure 7. It is noted that corrugated metal ducts in precast walls are not modelled in sectional analysis. This is primarily because: (i) there is typically no significant slip between the ducts and concrete, as the bond-slip typically occurs between the dowels and the grout inside the ducts [45–49], and (ii) pull-out failure of the ducts would only occur after extensive spalling and crushing of the concrete at the compression toe of the wall [14]. Regarding this latter point, the walls studied herein are typically governed by tension strains, while little, if any, concrete spalling of the wall boundaries is expected, which is consistent with the limited experimental testing available (i.e., [14]).

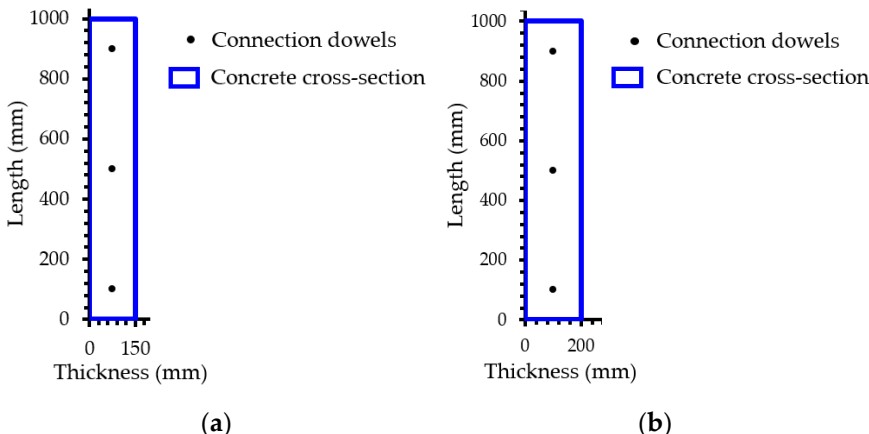

**Figure 7.** Definition of the critical cross-sections of SW1 and SW4 in WHAM [41]. (**a**) Cross-section of SW1 at base. (**b**) Cross-section of SW4 at base.

Step 3—Defining material properties: for assessment purposes, the measured uniaxial material properties (i.e., the stress–strain relationship) can be adopted in non-linear sectional analysis of precast walls. Therefore, the values measured in the experiment [14] for specimens SW1 and SW4 are adopted, as summarised in Table 2. Since the cracking of grout typically occurs in a very small lateral drift level, the grout bedding layer is not modelled [14]. For design purposes, the mean material properties, instead of the characteristic properties [20], are commonly required by the design code. For example, the commentary on Australian concrete design code AS 3600:2018 Sup1 [50] specifies Equation (6) for estimating the mean compressive strength of concrete ($f_{cmi}$). For the reinforcement, the mean tensile strains and stresses of different reinforcement grades typically used in Australia were tested [51,52]. For non-linear analysis, the mean yield and ultimate strength of D500N bars (normal-ductile deformed rebars in Australia) are taken as 550 MPa and 660 MPa, respectively [51]. The mean ultimate strain of D500N bars is approximately 9.5% [51]. For low-ductile D500L bars, the mean yield strength, ultimate strength, and ultimate strain are approximately 585 MPa, 620 MPa and 3.3%, respectively [51]. It should be noted that the mean material properties of concrete and reinforcement used in other countries and regions should be determined based either on experimental measurement or relevant design codes.

$$f_{cmi} = 0.9 f_{cm} = 0.9 \times \left(1.2875 - 0.001875 f_c'\right) f_c' \tag{6}$$

where $f_c'$ is the characteristic compressive strength of concrete; $f_{cm}$ is the mean value of cylinder strength.

**Table 2.** Measured material properties of SW1 and SW4 [14].

| Specimen No. | Concrete Compressive Strength | Connection Reinforcement | | |
| --- | --- | --- | --- | --- |
| | | Yield Strength | Ultimate Strength | Ultimate Strain |
| SW1 | 46 MPa | 473 MPa | 632 MPa | 10% |
| SW4 | 56 MPa | 473 MPa | 632 MPa | 10% |

Step 4—Defining the constitutive models of concrete and reinforcement and the tension stiffening model: an example of the material constitutive models used for moment–curvature analysis of lightly reinforced precast walls is illustrated in Figure 8. To be more specific, since the lightly reinforced precast concrete walls in existing buildings are mainly unconfined, as shown in Figure 1, the unconfined Karthik and Mander [53] concrete stress–strain relation is adopted. The Karthik and Mander [53] unconfined concrete model typically provides reasonable and conservative moment–curvature results, particularly for concrete with a compressive strength greater than 50 MPa [26]. In addition, compared

with the widely used Mander et al. [54] confined and unconfined concrete model, the Karthik and Mander [53] model can better predict the post-peak concrete stress–strain relationship [53]. Therefore, the Karthik and Mander [53] model is adopted here for the non-linear sectional analysis of lightly reinforced precast walls. It is noted that most non-linear sectional analysis software (e.g., WHAM) allows users to input their own material models. Therefore, engineers may apply a more precise concrete model based on test data. The Priestley et al. [24] reinforcing steel stress–strain model can be employed to model the dowel or wall reinforcement. The value of the steel yield plateau strain ($\varepsilon_{sh}$) mainly depends on the supplied form of reinforcement (straight or coil) and could vary significantly during tensile tests [26]. However, as an approximation, the yield plateau strain of D500N bars (i.e., normal ductility bars used in Australia) is around 2.4%, while no yield plateau was observed in tests for D500L bars (i.e., low ductility bars) [39]; in the latter case, the yield plateau strain is approximately equal to the yield strain of steel.

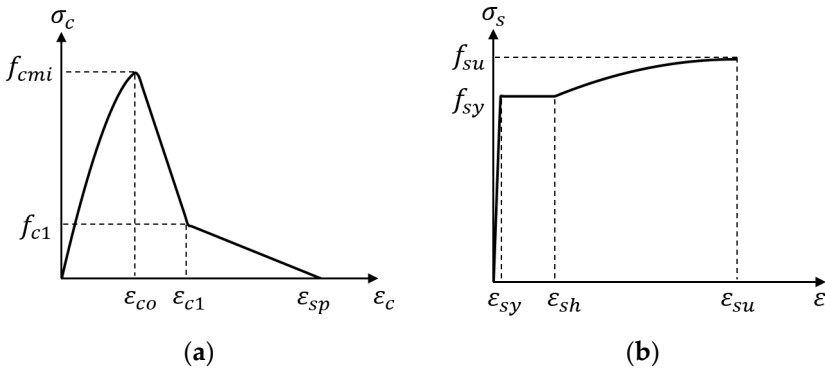

**Figure 8.** Material models for non-linear moment–curvature analysis of lightly reinforced precast concrete walls. (**a**) Karthik and Mander [53] unconfined concrete model. (**b**) Priestley et al. [24] steel model.

Another critical component of the non-linear sectional analysis of lightly reinforced precast walls is the modelling of tension stiffening, which could result in a significant difference between the local steel strain and the global average concrete strain [26,39,40,55]. Tension stiffening modelling is particularly crucial for lightly reinforced precast concrete walls for two primary reasons. Firstly, in lightly reinforced precast walls, the spacing between vertical reinforcement is typically much greater than in heavily reinforced walls, leading to more discrepancy between the local steel tensile strain and the average strain of the concrete section [40]. Secondly, the tension stiffening model is directly related to the bond stress–slip relationship between reinforcement and concrete [39,55]. In lightly reinforced precast walls with grouted dowels, lateral deformation is mainly contributed to by slippage between the connection dowels and grout. Therefore, the tension stiffening model should be considered in this case. The tension stiffening model in WHAM (illustrated in Figure 9) is the one developed by Menegon et al. [55], which was validated by experimental tests on cast-in-place wall boundary elements under cyclic tension and compression loading [39,55,56]. This specific tension stiffening model [55] has also shown promising results in comparison to three experimental precast wall boundary units, one of which was detailed with a low connection reinforcement ratio (in comparison to the detailing of the unit) [56].

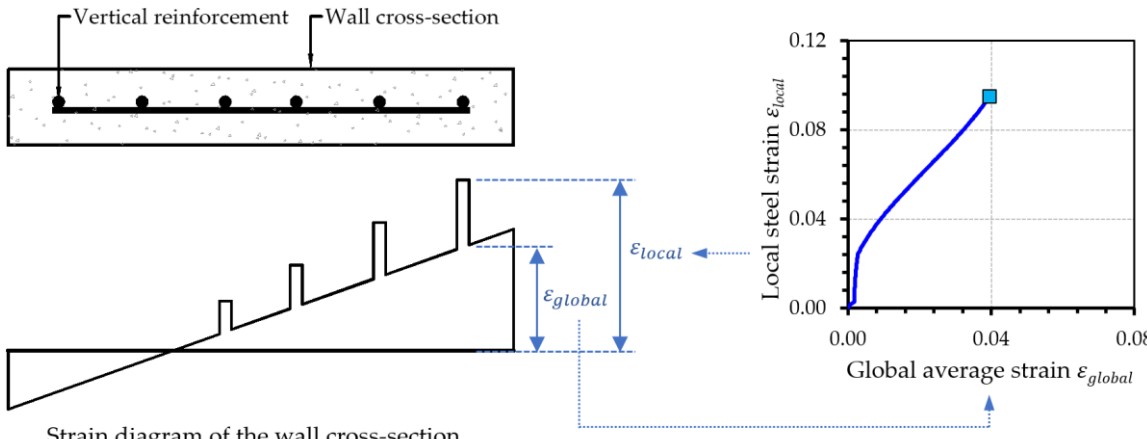

**Figure 9.** Illustration of the tension stiffening model proposed by Menegon et al. [55] (the figure is adapted from Menegon [39]).

Step 5—Defining the performance criteria and strain limits of concrete and reinforcement: a typical moment–curvature curve and its bi-linear approximation are illustrated in Figure 10. The authors acknowledge that more sophisticated envelopes of the moment–curvature (or force–displacement) response can be calculated by considering more refined models (e.g., the detailed wall model in Wibowo et al. [29] considering the cracking moment). However, for the purposes and aims of this study, a bi-linear approximation suffices. The governing points of the moment–curvature curve are computed based on the concrete or reinforcement strains adopted by Priestley et al. [24]. Table 3 summarises these strain limits, which correspond to different performance points. While the concrete strains are provided in Table 3, it is emphasised that, for the lightly reinforced precast walls that are the focus of this study, the tensile strains of the reinforcing steel will typically govern performance. The first yield point is defined as either the concrete reaching the compressive strain ($\varepsilon_{co}$) corresponding to the peak concrete strength (e.g., $f_{cmi}$) or the reinforcement reaching the yield strength ($\varepsilon_{sy}$), whichever is attained first [24]. The peak strain ($\varepsilon_{co}$) can be estimated based on the adopted concrete constitutive model, such as the Karthik and Mander model [53]. The nominal moment capacity ($M_{ny}$) is attained if the unconfined concrete strain exceeds 0.003 [26] or the steel tensile strain reaches 0.015 [24]. The corresponding nominal curvature ($\varnothing_{ny}$) is calculated by Equation (7) [24] based on the first yield curvature ($\varnothing'_y$) and first yield moment ($M'_y$). The ultimate moment–curvature capacity of the lightly reinforced precast walls studied here is deemed to occur when steel tensile strain of $0.6\varepsilon_{su}$ (where $\varepsilon_{su}$ is the ultimate tensile strain of steel) or concrete compressive strain of 0.004 is attained, whichever is exceeded first. Regarding the former, a reduced ultimate tensile strain value is used due to the low-cycle fatigue of reinforcement under reversed cyclic loads [24]. It is noted that a lower tensile strain limit of 0.04 has been recommended to prevent the occurrence of local rebar buckling [26]. However, the corrugated metal ducts in precast panels have been found to be able to restrain the dowels and help avoid bar buckling [56]. Therefore, for lightly reinforced precast walls that have their critical cross-sections at the interface between the walls and the foundation (e.g., Figure 6), a steel tensile strain of $0.6\varepsilon_{su}$ can be adopted for the ultimate point in the moment–curvature curve. An unconfined concrete strain of 0.004 is adopted from Priestley et al. [24]; in the case where a confined ductile wall is analysed (outside of the scope of this paper), a larger ultimate concrete compressive strain value can be used, which can be calculated, for example, using the equations proposed in Mander et al. [54]. The ultimate moment of the bi-linear curve ($M_{bu}$) is taken as the maximum moment ($M_{max}$) obtained from the non-linear sectional analysis.

$$\varnothing_{ny} = \frac{M_{ny}}{M'_y}\varnothing'_y \qquad (7)$$

$$M_{bu} = M_{max} \qquad (8)$$

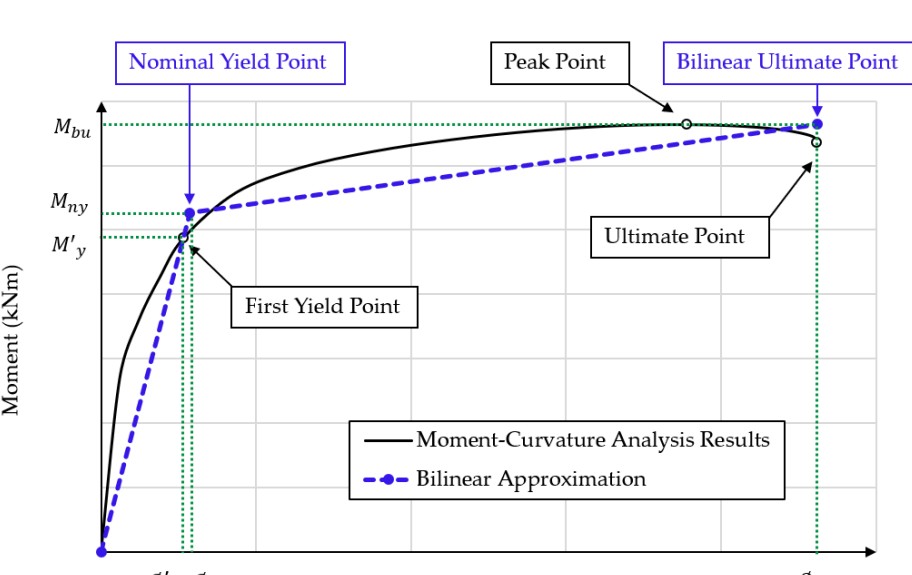

**Figure 10.** A typical moment–curvature curve and bi-linear approximation.

**Table 3.** Material strain limits for moment–curvature analyses of lightly reinforced load-bearing precast walls.

|  | First Yield | Nominal Yield | Ultimate Limit State |
|---|---|---|---|
| Concrete Compressive Strain ($\varepsilon_c$) | $\varepsilon_{co}$ | 0.003 | 0.004 |
| Reinforcement Tensile Strain ($\varepsilon_s$) | $\varepsilon_{sy}$ | 0.015 | $0.6\varepsilon_{su}$ |

Step 6—Computation of the moment and curvature values: Lam et al. [35] provide a comprehensive explanation of the principles of and procedures for conducting a moment–curvature analysis using an EXCEL spreadsheet. The WHAM program [41] used in this study essentially follows the same methodology. In summary, the computational workflow includes the following:

1. Subdividing the cross-section into several rectangular concrete splices with equal depth.
2. Defining a tentative (i.e., unbalanced) global reference strain (i.e., concrete strain) at the extreme concrete fibre and assuming a neutral axis.
3. Computing the strains of other concrete splices and converting the global average concrete strain into the local steel strain using the defined tension stiffening model.
4. Computing the stresses and forces of each concrete splice and reinforcing steel bar based on the selected material models. The tensile stresses (i.e., tensile strength) of concrete are typically not considered in sectional analysis.
5. By iterative computation, finding out the accurate depth of the neutral axis and the balanced reference strain via force equilibrium of the internal forces (i.e., forces of each concrete splice and rebar) and the external axial load.
6. Computing the moment and curvature results of the balanced cross-section. The moment can be calculated from the balanced forces, and the curvature can be estimated as the ratio of the balanced reference strain and the depth of the neutral axis.
7. By repeating the above processes for gradually increasing reference strains and reaching each governing point (i.e., the yield and ultimate points discussed above), the moment and curvature relationship of a precast RC wall section can be plotted. The curved moment–curvature relationship can be converted to a bi-linear approximation based on the approach shown in Figure 10.

8. If the cross-section is not rectangular, the most critical moment–curvature scenario should be identified by considering different concrete compression faces.

By adopting the moment–curvature analysis proposed above, the bi-linear moment–curvature capacities of the two wall units SW1 and SW4 [14] are computed, as summarised in Table 4.

**Table 4.** Bi-linear moment–curvature analyses results for SW1 and SW4.

| | Nominal Yield | | Ultimate | |
|---|---|---|---|---|
| | Curvature (1/km) | Moment (kNm) | Curvature (1/km) | Moment (kNm) |
| SW1 | 1.9 | 125.4 | 52.7 | 171.4 |
| SW4 | 1.2 | 123.6 | 35.3 | 178.2 |

*2.3. Plastic Hinge Analysis of Rectangular Lightly Reinforced Precast Walls*

A simple approach to evaluate the lateral force–displacement response of lightly reinforced precast walls is plastic hinge analysis (PHA) [24]. An essential and critical component of PHA for RC structural walls is the prediction of the plastic hinge length (PHL) at the wall base. Several past research studies have proposed equations for predicting the PHL of cast-in-situ walls with varying detailing [24,57–61]. One of the most recent expressions, developed by Hoult [59], discusses how these expressions can provide a large range of values, which are highly dependent on the limiting design values used to derive the expressions. However, it would be inappropriate to use these expressions, which were specifically developed for cast-in-situ RC walls, for the purposes of assessing the global seismic performance of lightly reinforced precast walls, where it is assumed that plasticity is concentrated at a single base crack.

The fundamentals of plastic deformation in lightly reinforced precast walls are expected to be similar to those of conventional lightly reinforced cast-in-situ walls (e.g., [57]), which is largely due to the strain penetration of the longitudinal rebars. A detailed review and explanation of the strain penetration mechanism in lightly reinforced cast-in-situ walls is provided by Altheeb [62]. However, the primary source of plastic deformation in lightly reinforced precast walls is a combination of the strain penetration of the dowels into the foundation and that into the coregulated metal tubes (in the wall) [29,56,63]. The New Zealand guidelines [64] specify Equation (9) to estimate the plastic hinge length of lightly reinforced precast walls with grouted dowels (and an assumed single crack forming at the base). The strain penetration length ($L_{sp}$) in Equation (9) is that adopted from Paulay and Priestley [65] but (conservatively) multiplied by a factor of 1.5 to account for strain penetration into two sides (above and below) of the single primary crack.

$$L_p = 1.5L_{sp} = 1.5 \times 0.022 f_{sy} d_{dowel} \tag{9}$$

The New Zealand guidelines [64] explain that a reduced length of $0.5L_{sp}$ is expected for the strain penetration of dowels into ducts of precast walls because it is believed that the presence of these ducts could help reduce overall vertical elongation (i.e., anchorage slip). However, recent experimental tests on precast wall boundary elements found that the reduced post-peak bond strength of the grout in the ducts, in comparison to typical concrete, causes more bar slip [56]. This degraded inelastic bond between reinforcing dowels and grout was also observed by other researchers in small-scale pull-out tests of grout tube connections [46]. Based on this limited literature evidence, the authors believe that the adoption of a plastic hinge length of $1.5L_{sp}$ might be too conservative for the purposes of seismic assessment of lightly reinforced precast walls with dowelled connections.

Consequently, for seismic assessment purposes, an equivalent PHL of $L_p = 2L_{sp}$, as shown in Equation (10), is recommended to evaluate rectangular lightly reinforced precast walls, which assumes that performance is controlled by strain penetration of the reinforcing steel dowels (i.e., the connection reinforcement) into the foundation and the

ducts, as illustrated in Figure 11. It is noted that increasing the axial loads applied to precast walls would help mobilise the development of flexural deformation, in which case an enlarged PHL can be expected. However, due to the paucity of experimental and numerical studies on these insufficiently detailed precast walls, it is still conservative to adopt the recommended PHL of $2L_{sp}$ if the precast RC walls have similar detailing to the walls discussed in this paper. Some numerical investigations are currently being undertaken by the authors to study the localised bond–slip behaviour of the dowel reinforcement in grouted ducts commonly used in precast walls. It is expected that a more precise tension stiffening model and strain penetration length for dowels in typical grouted duct connections can be developed based on these numerical investigations. Subsequently, the recommendations for the equivalent PHL here will be reviewed.

$$L_p = 2L_{sp} = 0.044 f_{sy} d_{dowel} \tag{10}$$

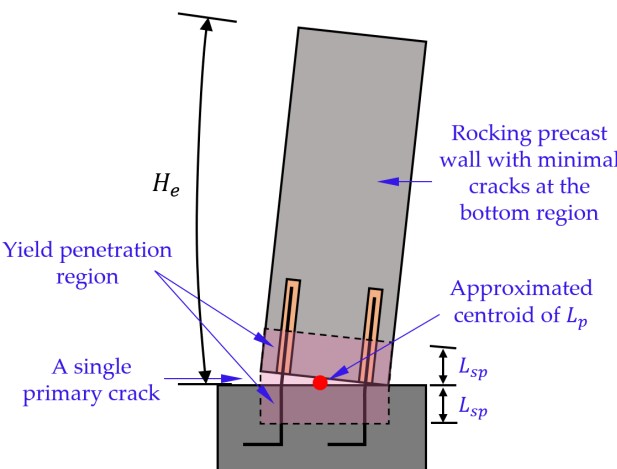

**Figure 11.** The phenomenon of yield penetration and plastic deformation.

Based on the moment–curvature results (i.e., $\varnothing_{ny}$, $M_{ny}$, $\varnothing_u$ and $M_{bu}$ in Section 2.2), the bi-linear force–displacement capacity of a lightly reinforced load-bearing precast wall can be estimated using Equations (11)–(17), which were adopted by Priestley et al. [24]. The yield strength ($F_{y,B}$) and ultimate strength ($F_{u,B}$) of a building supported by *n* number of rectangular precast walls is the summation of each individual yield ($F_{y,W}$) and ultimate capacity ($F_{u,W}$), respectively, of the precast walls with their major axis perpendicular to the loading direction if floor slabs are considered rigid. Otherwise, a more rigorous structural analysis (e.g., finite element analysis) is required to evaluate the effect of diaphragm flexibility on the distribution of lateral forces and overall structural behaviour. The ultimate displacement capacity of the building is conservatively taken as the lowest ultimate capacity value ($\Delta_{eu}$) of any of these contributing precast structural walls.

$$\Delta_{ey} = \frac{\varnothing_{ny} H_e^2}{3} \tag{11}$$

$$F_{y,W} = \frac{M_{ny}}{H_e} \tag{12}$$

$$F_{y,B} = \sum F_{y,W} \tag{13}$$

$$\Delta_{ep} = (\varnothing_u - \varnothing_y) L_p H_e \tag{14}$$

$$\Delta_{eu} = \Delta_{ey} + \Delta_{ep} \tag{15}$$

$$F_{u,W} = \frac{M_{bu}}{H_e} \tag{16}$$

$$F_{u,B} = \sum F_{u,W} \tag{17}$$

where $\Delta_{ey}$, $\Delta_{ep}$ and $\Delta_{eu}$ are the effective yield, plastic, and ultimate displacement capacities, respectively, of the equivalent SDOF precast wall. The effective height ($H_e$) of the equivalent SDOF precast wall system is illustrated in Figure 3a and can be taken as 70% of the full height of the wall (i.e., $0.7H_n$, where $H_n$ is the total height of the building) [24].

The lateral drift profile of the MDOF wall can be estimated using Equation (18), by modifying the expressions proposed by Lam et al. [35] based on Priestley et al. [24].

$$\Delta_{u,i} = \Delta_{y,i} + \Delta_{p,i} = \frac{3}{2}\Delta_{ey}\left(\frac{h_i}{H_e}\right)^2\left(1 - \frac{h_i}{3H_n}\right) + \Delta_{ep}\frac{h_i}{H_e} \tag{18}$$

where $\Delta_{y,i}$, $\Delta_{p,i}$ and $\Delta_{u,i}$ are the yield, plastic, and ultimate displacement, respectively, of a building height of $h_i$ above the foundation.

By adopting the proposed PHA approach, the bi-linear force–displacement capacity curves of SW1 and SW4 are calculated based on the moment–curvature results provided in Table 4. The capacity curves obtained from PHA are plotted in Figure 12 and are compared with experimental reversed-cyclic force–displacement curves [14]. The testing data can be accessed through the dataset provided by Seifi et al. [66]. From the comparison, the PHA methods used in this research provide a very reasonable prediction of the force–deformation response of wall unit SW1. For SW4, the PHA method underestimates the ultimate displacement of the precast wall in comparison to the experimentally attained displacement capacity. The underestimation of the displacement capacity using the PHA undertaken here is likely to be due to a lower PHL ($L_p$), where greater strain penetration was possibly achieved experimentally in comparison to that calculated here using Equation (10). As measured in experiment [14], approximately 92% of the deformation contribution to SW4 was due to rocking, where the behaviour of the wall would be controlled by dowel anchorage slip into the foundation and dowel bar slip into the ducts of the wall. There was also a small contribution to the overall displacement of SW4 due to sliding [14], which was not included in the PHA undertaken here. Overall, the calculated force–displacement curves from the PHA method recommended here were found to be reasonable, albeit slightly conservative in one instance (i.e., SW4), in comparison to the paucity of experimental results available.

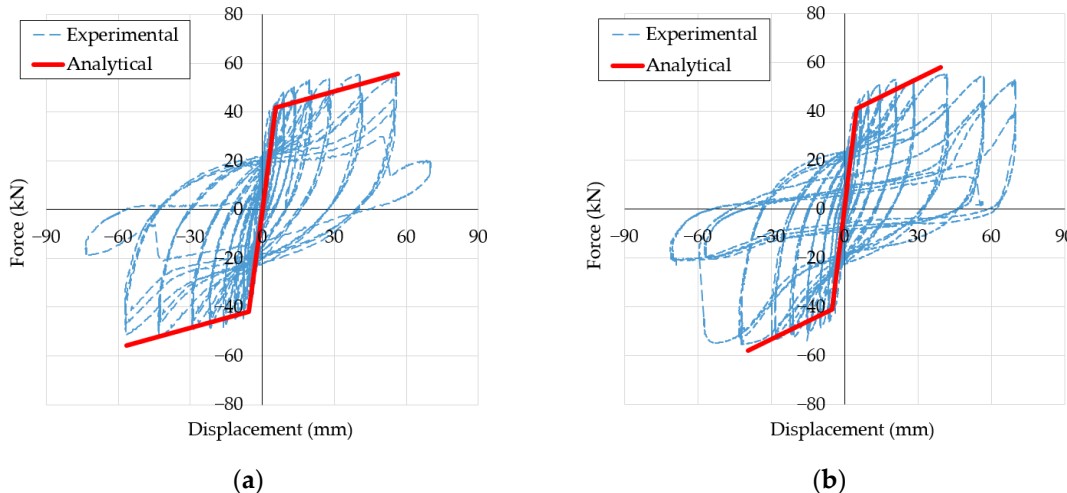

**Figure 12.** Plastic hinge analysis results for wall units SW1 and SW4 [14]. (**a**) SW1. (**b**) SW4.

## 3. Simplified Push-Over Analysis of a Case Study Building Supported by Precast Walls

A six-storey case study building supported by four rectangular lightly reinforced precast walls, as shown in Figure 13a, is evaluated using the proposed simplified push-over analysis method as a demonstration. The building layout is similar to the case study building (CSB1) used in Khatiwada et al. [6], but all the lateral-load-resisting structural

walls here are assumed to be lightly reinforced precast walls, as shown in Figure 13b. The floor system is assumed to consist of 200 mm thick flat slabs, which are commonly employed in Australia for both cast-in-situ and precast buildings [12,17]. It is assumed that these floor slabs have sufficient stiffness to behave as rigid diaphragms in distributing lateral loads to vertical structural elements. The vertical connections (if any) between precast panels are assumed to be wet joints with sufficient rigidity to develop the full composite capacity of the connected walls.

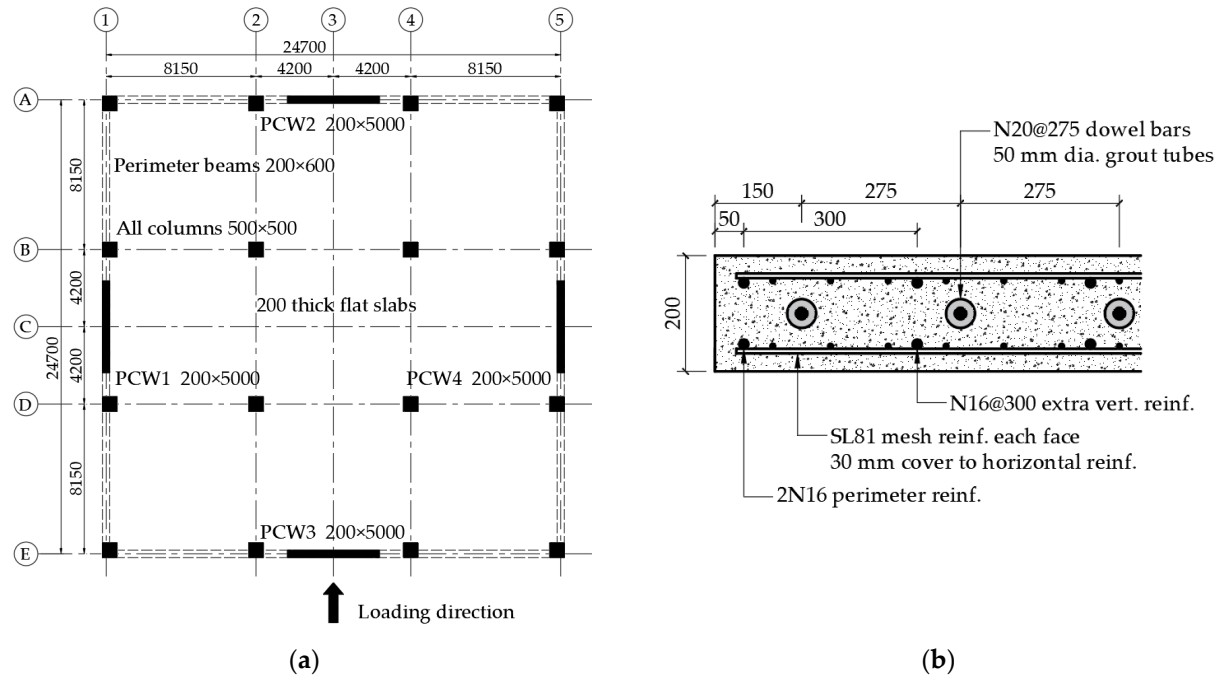

**Figure 13.** Case study building supported by rectangular lightly reinforced precast walls. (**a**) Typical floor plan of the case study building. (**b**) Cross-section of precast walls at the base.

The design information of the case-study building is summarised in Table 5. The axial load ratio of 5% is determined from the tributary area approach and it is assumed that the axial load acts at the centroid of the wall section. The load is calculated by considering the self-weight of structural elements and floor systems, a superimposed dead load of 2 kPa, and a live load (i.e., imposed actions) of 2 kPa.

**Table 5.** Parameters of the case study building.

| Storey | Total Height | Height (Ground Floor) | Height (Other Floors) | Total Building Mass | Axial Load Ratio |
|--------|-------------|----------------------|----------------------|--------------------|--------------------|
| 6 | 19.3 m | 3.8 m | 3.1 m | 3700 tons | 5% |

Moment–curvature analysis of the case study building is performed using WHAM [39–41] following the steps and principles specified in Section 2.2 of this paper. The mean in-situ strength of the concrete is 53.7 MPa. The mean yield strength of the connection dowels is 550 MPa, and the mean ultimate steel strength and strain are 660 MPa and 9.5%, respectively. The longitudinal reinforcement ratio of the precast panels is assumed to be 0.84%, which is greater than the connection reinforcement ratio of 0.57%. The strain limits for the corresponding performance points used in sectional analysis are summarised in Table 6. The calculated bi-linear moment–curvature results of PCW1, which include the points at the nominal yield and ultimate, are presented in Table 7. Since PCW1 and PCW4 are assumed to have the same dimensions and detailing, the curvature and moment capacities of PCW4 are assumed to be the same as those of PCW1.

**Table 6.** Material strain limits for moment–curvature analyses of the case study precast walls.

| | First Yield | Nominal Yield | Ultimate Limit State |
|---|---|---|---|
| Reinforcement Tensile Strain ($\varepsilon_s$) | 0.00275 | 0.015 | $0.6\varepsilon_{su} = 0.057$ |

**Table 7.** Bi-linear moment–curvature results of PCW1.

| | Nominal Yield | Ultimate |
|---|---|---|
| Curvature (1/km) | 0.58 | 4.62 |
| Moment (kNm) | 10,522 | 13,272 |

The bi-linear force–displacement capacity curve of the case study building is assessed as follows, based on the PHA approach specified in Section 2.3. The effective height and effective mass are estimated to be 70% of the total building height and mass, respectively [24]. Because the lateral drift response of these lightly reinforced precast walls is controlled by rocking action, the plastic hinge length ($L_p$) is taken as $2L_{sp}$, as specified in Section 2.3. The total force capacity of the building is the summation of the strength of the precast walls (i.e., PCW1 + PCW4). Since PCW1 and PCW4 are assumed to have identical detailing, the ultimate deformation capacity of the building is taken to be the ultimate displacement of PCW1 obtained via PHA.

$$\Delta_{ey} = \frac{\varnothing_y H_e^2}{3} = 35 \text{ mm} \tag{19}$$

$$F_{y,PCW1} = \frac{M_{ny}}{H_e} = 779 \text{ kN} \tag{20}$$

$$F_{y,B} = 2 \times F_{y,PCW1} = 1558 \text{ kN} \tag{21}$$

$$L_p = 2L_{sp} = 484 \text{ mm} \tag{22}$$

$$\Delta_{eu} = \Delta_{ey} + \Delta_{ep} = \frac{\varnothing_y H_e^2}{3} + (\varnothing_u - \varnothing_y) L_p H_e = 62 \text{ mm} \tag{23}$$

$$F_{u,PCW1} = \frac{M_{bu}}{H_e} = 982 \text{ kN} \tag{24}$$

$$F_{u,B} = 2 \times F_{u,PCW1} = 1964 \text{ kN} \tag{25}$$

The bi-linear capacity curve (red solid line) of the case study building supported by precast walls, in the acceleration and displacement format with the controlling points calculated above, is plotted in Figure 14. The inelastic site-specific response spectra for four reference periods (0.2 s, 0.5 s, 1.0 s and 2.0 s, respectively) superimposed on this same figure are those developed by Khatiwada et al. [6] on site class D$_e$ (defined in AS 1170.4 [1]) under a return period of 2500 years. Readers can refer to Khatiwada et al. [6] for derivations of these site-specific spectra. Similarly, the response spectra for other building sites can be derived based on the procedures proposed in previous studies [3,5,6] if geotechnical reports (e.g., borehole records) on the sites are available. The inelastic response spectra are converted from the elastic response spectra using Equations (3)–(5). The overstrength factor ($\Omega$) is taken to be 1.30 for lightly reinforced (i.e., non-ductile) precast walls per AS 1170.4 [1]. The ductility factor ($\mu$) is approximately 1.75, calculated as the ratio of the ultimate displacement (62 mm) and yield displacement (35 mm). The calculated ductility factor is greater than the value of 1.0 prescribed for non-ductile precast walls in AS 1170.4 [1], but smaller than the value of 2.0 corresponding to 'limited-ductile' walls. For comparison, the inelastic code

response spectrum on site class $D_e$ [1] under a 2500-year return period is also plotted in Figure 14. Similarly, Equations (3)–(5) are employed to convert the elastic code response spectrum specified in AS 1170.4 [1] for a hazard design factor (Z) of 0.08 (corresponding to a peak ground acceleration of 0.08 g for a return period of 500 years) and a probability factor ($K_p$) of 1.8 (used to amplify the hazard design factor to a 2500-year return period event) to the inelastic code response spectrum shown in the figure.

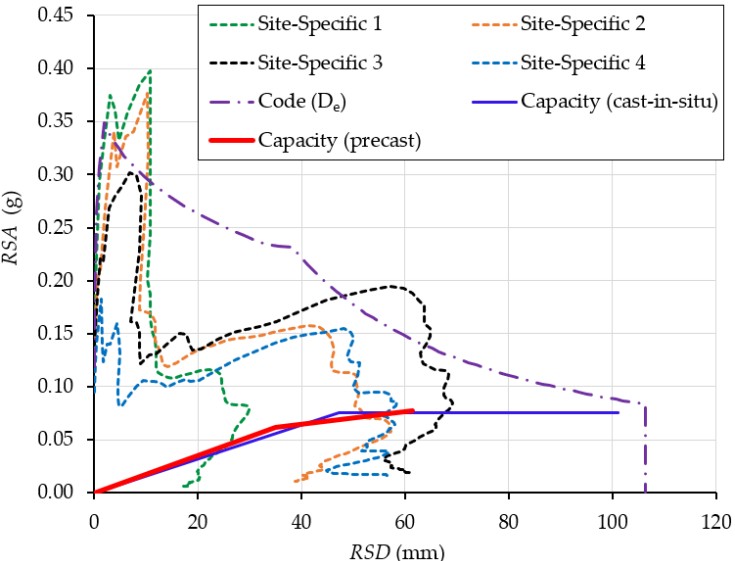

**Figure 14.** Capacity and demand curves of the case study building. The code demand curve was derived assuming a 2500-year return period ($K_pZ = 0.144$ g) on site class $D_e$.

According to the analytical results, the ultimate deformation capacity of the case study building is within the envelope of site-specific response spectrum 3, which means an unsatisfactory force–displacement capacity of the building, thus indicating seismic vulnerability of these lightly reinforced precast concrete walls in this building site. In order to enhance the deformation capacity of precast walls, it is recommended to improve the detailing practices by providing a sufficient amount of connection dowels (at least greater than the wall's vertical reinforcement content) and enough confinement reinforcement in precast panels as well as around grouted dowel connections, such as following the recent amendments in AS 3600:2018 [20] and AS 3600:2018 Sup1 [50]. If satisfactory detailing is adopted and sufficient ductility is attained in precast walls, the lateral drift response of the precast walls would be governed by the flexural-shear failure mechanism, similar to conventional limited-ductile cast-in-situ walls [67] and, thus, achieve better plastic deformation capacity. It is noted that some concrete structure codes (e.g., AS 3600:2018 [20]) require lightly reinforced precast walls to be designed as non-ductile using full elastic earthquake actions. In this case, non-linear analysis should only be used for seismic assessment purposes, but not for design purposes.

Furthermore, in order to compare the seismic performance of lightly reinforced precast walls and limited-ductile cast-in-situ walls, Figure 14 presents the force–displacement capacity of a building laterally supported by limited-ductile cast-in-situ RC walls with the same structural layout, material properties and longitudinal reinforcement ratio (0.57%) to the case study precast building. Calculations of the ultimate force and deformation of the limited-ductile RC wall are provided in Appendix A, based on the expressions adopted by Khatiwada et al. [6]. It can be observed from Figure 14 that the ultimate forces of the two buildings are almost identical. However, the ultimate displacement of the building laterally supported by precast walls is significantly lower than that of the cast-in-situ building due to the significantly reduced plastic hinge length of these lightly reinforced precast walls. This discrepancy implies that if seismic design is only based on the strength of

lateral-load-resisting elements, the seismic vulnerability of such precast structural elements, in terms of displacement demands, will be overlooked.

According to the results shown in Figure 14, there is a noticeable discrepancy between the code spectrum and the site-specific response spectra. For a small or medium *RSD*, using the code spectrum may underestimate earthquake demands on the building, thus increasing the seismic vulnerability of the building. However, in most ranges of *RSD*, the code spectrum is typically over-conservative. For instance, the seismic performance of a building laterally supported by cast-in-situ walls can be regarded as satisfactory if the assessment is based on the site-specific response spectra, even though the ultimate capacity may be slightly within the code design curve. Hence, a site-specific response spectrum is also a good tool for design optimisation.

## 4. Conclusions

Precast reinforced concrete walls in many existing multi-storey buildings in low-to-moderate seismicity regions, such as Australia, are found to be commonly designed with unsatisfactory reinforcement detailing due to a lack of sufficient knowledge of the seismic behaviour of these precast elements. These precast walls are typically lightly reinforced with low-ductile mesh reinforcement and have a connection reinforcement ratio that is less than the wall reinforcement ratio. Recent experimental testing on these precast walls with grouted dowel base connections has identified their potential seismic deficiencies. Therefore, it is important to evaluate the seismic performance of these lightly reinforced precast walls. This paper presents a detailed framework for conducting seismic assessment of these precast walls by employing a displacement-based approach.

In this study, displacement-based seismic assessment is based on the widely used capacity spectrum method. The capacity spectrum method includes two main components, namely the capacity curve and the demand curves. This paper comprehensively explains the principles of and steps for predicting the capacity curve of rectangular lightly reinforced precast walls using non-linear sectional-based moment–curvature analysis and plastic hinge analysis. The equivalent plastic hinge length of the lightly reinforced precast walls, where the lateral response is dominated by rocking with the fracturing of the connection dowels governing performance, is proposed by considering the strain penetration of the dowels into the foundation and into the ducts. Some of the limited experimental testing on such precast systems is used to help validate the methods proposed herein. Inelastic site-specific demand curves, in the form of acceleration–displacement response spectra, corresponding to a 2500-year return period event in an Australian city are used to determine the seismic performance of a case study building that utilises lightly reinforced precast walls. The results of this case study highlight the potential seismic vulnerability of these precast elements in multi-storey buildings and also emphasise the importance of reinforcement detailing to achieve a ductile response of these salient lateral load-resisting elements. The proposed seismic analysis approach is applicable to lightly reinforced precast concrete walls in other countries and regions with similar detailing to that described in this paper.

**Author Contributions:** Conceptualisation, X.W., R.D.H. and E.L.; methodology, X.W., R.D.H. and E.L.; software, X.W.; validation, X.W., R.D.H. and E.L.; formal analysis, X.W.; investigation, X.W.; resources, R.D.H. and E.L.; writing—original draft preparation, X.W.; visualisation, X.W.; writing—review and editing, X.W., R.D.H. and E.L.; supervision, R.D.H. and E.L. All authors have read and agreed to the published version of the manuscript.

**Funding:** This research received no external funding.

**Institutional Review Board Statement:** Not applicable.

**Informed Consent Statement:** Not applicable.

**Data Availability Statement:** Not applicable.

**Acknowledgments:** This work was supported by the Melbourne Research Scholarship provided by the University of Melbourne.

**Conflicts of Interest:** The authors declare no conflict of interest.

## Appendix A. Prediction of the Force–Displacement Capacity of a Limited-Ductile Reinforced Concrete Wall

The force–displacement capacity of a limited-ductile reinforced concrete wall which has the same dimensions, material properties, and reinforcement ratio (0.57%) used in the case study building in this research (i.e., Figure 13 and Table 5) is estimated using the equations adopted by Khatiwada et al. [6] as follows.

$$\varnothing_y = \left(\frac{b_w L_w^3}{12 I_{gross}}\right)^{0.45} \frac{\left(0.15\rho_v - 2\rho_v^2 + 0.0031\right)}{L_w} = 7.77 \times 10^{-7} \text{ mm}^{-1} \tag{A1}$$

$$\varnothing_u = \left(\frac{b_w L_w^3}{12 I_{gross}}\right)^{0.45} \frac{\left(19.5\rho_v - 545\rho_v^2 - 0.066\right)(0.158 - ALR) + 0.017}{L_w} = 3.98 \times 10^{-6} \text{ mm}^{-1} \tag{A2}$$

$$E_c I_{eff} = E_c I_g [\rho_v(10 - 30 ALR) + 0.03 ALR f_{cmi} + 0.1] = 1.66 \times 10^{16} \text{ Nmm}^2 \tag{A3}$$

$$\Delta_y = \frac{\phi_y H_e^2}{3} = 47.3 \text{ mm} \tag{A4}$$

$$L_p = Min\left[0.2\left(f_{su}/f_{sy} - 1\right), 0.08\right] \times H_e + 0.1 L_w + 0.022 f_{sy} d_b = 1282.4 \text{ mm} \tag{A5}$$

$$\Delta_u = \Delta_y + \left(\phi_u - \phi_y\right) L_p \times \left(H_e - 0.5 L_p + L_{sp}\right) = 101.1 \text{ mm} \tag{A6}$$

$$F_{y,W} = F_{u,W} = \frac{E_c I_{eff} \phi_y}{H_e} = 953 \text{ kN} \tag{A7}$$

$$F_{y,B} = F_{u,B} = 2 \times F_{u,W} = 1906 \text{ kN} \tag{A8}$$

where $b_w$ and $L_w$ are the thickness and length of the wall, respectively; $I_{gross}$ and $I_{eff}$ are the gross and effective second moment of area, respectively; $\rho_v$ is the longitudinal reinforcement ratio of the wall; $ALR$ is the axial load ratio of the wall; $E_c$ is the modulus of elasticity of concrete.

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
