# Peer review of "Site-Specific Seismic Analysis of Buildings Supported by Lightly Reinforced Precast Concrete Walls"

_2673-4109, doi:10.3390/civileng4010016_

Round 1
Reviewer 1 Report
This paper presents seismic analyses for buildings with lightly reinforced precast concrete walls. Pushover analyses superimposing on inelastic site-specific response spectrum were conducted. The detail procedure for the analysis was presented. The presented research is suitable for the Journal. The reviewer would like to provide the following comments:
1. Line 13. “connection dowel reinforcement ratio”. Please avoid multiple nouns before a noun.
2. Line 15. How about the shear friction behavior? It is not discussed in the paper.
3. Line 33. Does this limitation only apply to the AS1170? How about other codes: ASCE-7, EC-8, and NZS etc.
4. Line 63. It would be better to show typical reinforced ratio and maximum spacing in the longitudinal reinforcement.
5. Line 68. “effort” to “efforts”.
6. Line 76. Is the floor slab precast? If so, can it be considered as rigid?
7. Line 78. If in the design assumption the lateral load will not be transferred to the PC walls, what is the lateral force resisting system to carry it?
8. Line 172. It is better to use different line types/colors for plot (b).
9. Line 279. Please indicate the reinforcement ratio in this Table.
10. Line 309. Please annotate this figure for a better understanding. For example, the units of axis, what the black dots and orange dashed line stand for.
11. Line 363. Please illustrate the tension-stiffening model with a schematic.
12. Line 579. Should it be site-specific response spectrum 3? How about other sites? What are the differences between each site?
13. Line 579. Also, please discuss the code design curve. Does it provide a more conservative design than other site-specific curves?
14. Line 605. Please place the capacity curve to Fig. 13.
Author Response
Dear Reviewer,
The authors sincerely appreciate your comments. Please see the attachment for the authors' response.

Reviewer 2 Report
In this research, a study was carried out on the application of the site-specific response spectra in the analysis of buildings that are supported by lightly reinforced precast concrete walls. This paper uses moment curvature analyses in combination with plastic hinge analyses to evaluate the force-displacement capacity of planar lightly reinforced load-bearing precast walls. The seismic performance of a building supported was assessed by superimposing the demand curve and the inelastic site-specific response spectra developed for the building site. This is a very important topic and it fits well with the scope of the journal. From my point of view, as a researcher in materials science, I do not find any problem in accepting this research as it is.
Author Response
Dear reviewer,
The authors sincerely appreciate your acceptance of this manuscript. Thank you.